# Determining spin-orbit coupling in graphene by quasiparticle interference imaging

Lihuan Sun [1], Louk Rademaker[1,2], Diego Mauro[1,3], Alessandro Scarfato [1], Árpád Pásztor [1], Ignacio Gutiérrez-Lezama [1,3], Zhe Wang [1,4], Jose Martinez-Castro [1], Alberto F. Morpurgo [1,3] & Christoph Renner [1] ✉

Inducing and controlling spin-orbit coupling (SOC) in graphene is key to create topological states of matter, and for the realization of spintronic devices. Placing graphene onto a transition metal dichalcogenide is currently the most successful strategy to achieve this goal, but there is no consensus as to the nature and the magnitude of the induced SOC. Here, we show that the presence of backscattering in graphene-on-WSe$_2$ heterostructures can be used to probe SOC and to determine its strength quantitatively, by imaging quasiparticle interference with a scanning tunneling microscope. A detailed theoretical analysis of the Fourier transform of quasiparticle interference images reveals that the induced SOC consists of a valley-Zeeman ($\lambda_{vZ} \approx 2$ meV) and a Rashba ($\lambda_R \approx 15$ meV) term, one order of magnitude larger than what theory predicts, but in excellent agreement with earlier transport experiments. The validity of our analysis is confirmed by measurements on a 30 degree twist angle heterostructure that exhibits no backscattering, as expected from symmetry considerations. Our results demonstrate a viable strategy to determine SOC quantitatively by imaging quasiparticle interference.

Phase coherent propagation of electrons in solids is a basic quantum mechanical process that determines the low-temperature transport properties of metallic conductors[1–4]. Key in this context is the notion of backscattering—the ability of electrons to precisely reverse their direction of propagation as a result of elastic collisions—because how electrons backscatter is deeply related to the structure of the electronic wavefunctions and to the symmetries present in the system[3]. For electrons described by scalar wavefunctions in conventional metals, for instance, time reversal symmetry causes constructive interference that enhances backscattering and suppresses conductivity, resulting in the well-known phenomenon of weak-localization[1,3,4]. In the presence of SOC—for which the spinorial structure of wavefunctions needs to be considered—time reversal symmetry instead results in destructive interference that suppresses (rather than enhancing) backscattering, leading to weak antilocalization[2,3].

For Dirac electrons in graphene—i.e., for electrons that propagate while remaining in the same valley—the situation is opposite to that of conventional electrons[5–8]. The difference is rooted in the two-component, pseudo-spinorial wavefunction of Dirac electrons, which originates from the two inequivalent atoms forming the graphene unit cell[9,10]. The Hamiltonian of pseudo-spinorial Dirac fermions in graphene possesses an additional symmetry—so-called chiral symmetry[9]—which ensures that electrons with opposite momenta have orthogonal pseudospins (if time reversal symmetry is also present; Fig. 1a), resulting in the complete suppression of backscattering[5,11,12]. However, if SOC is strong enough, two more components need to be added to the wavefunction. The resulting four-component wavefunctions of electrons propagating in opposite directions are no longer orthogonal (Fig. 1b), and backscattering is again possible. It follows that the ability to detect whether or not Dirac electrons in monolayer graphene can

[1]Department of Quantum Matter Physics, University of Geneva, 1211 Geneva, Switzerland. [2]Department of Theoretical Physics, University of Geneva, 1211 Geneva, Switzerland. [3]Group of Applied Physics, University of Geneva, 1211 Geneva, Switzerland. [4]MOE Key Laboratory for Nonequilibrium Synthesis and Modulation of Condensed Matter, Shaanxi Province Key Laboratory of Advanced Materials and Mesoscopic Physics, School of Physics, Xi'an Jiaotong University, 710049 Xi'an, China. ✉e-mail: christoph.renner@unige.ch

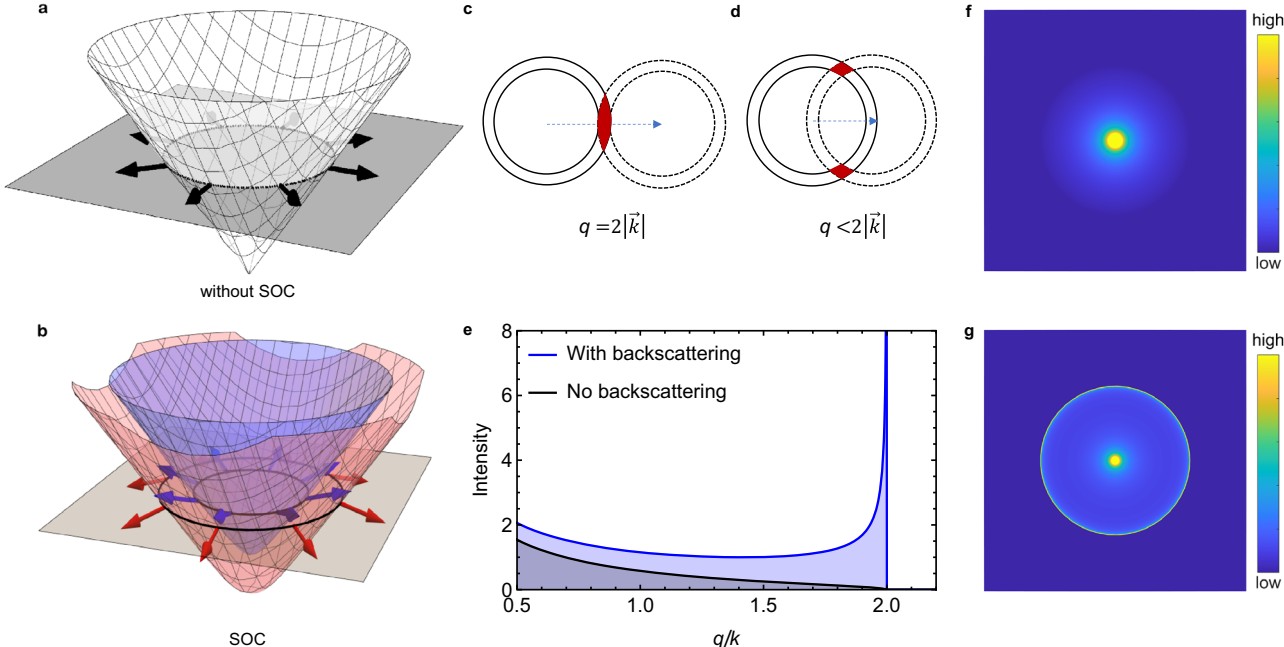

**Fig. 1 | Pseudospin winding in graphene and QPI patterns with and without backscattering. a** Pseudospin winding (black arrows) in a single Dirac cone. **b** Pseudospin winding and spin-orbit split bands of graphene in the presence of SOC. Blue and red arrows illustrate the tilted pseudospin winding. The blue (red) Dirac cone corresponds to spin up (down) configuration. **c, d** The geometrical constructions of the phase overlap of electron scattering at a constant energy. The cross section of incoming $\vec{k}$ and scattered $\vec{k}'$ states are marked by filled red areas, where (**c**) depicts backscattering. To model the suppression of backscattering, we include angle-dependent scattering of the form $\cos^2(\theta/2)$. **e** The scattering electron phase overlap as a function of scattering momentum $q = |\vec{k} - \vec{k}'|$ if backscattering is allowed (blue curve) or forbidden (black curve). The absence of any scattering at momenta beyond $2k$ is an artefact of this approximate geometric construction. **f** Phenomenological QPI power spectrum without and **g** with backscattering.

backscatter provides a unique probe of the presence and strength of SOC.

Here, we establish the occurrence of backscattering of Dirac electrons in single-layer graphene (SLG)-on-WSe$_2$ by analysing quasiparticle interference images obtained by means of large-area scanning tunneling spectroscopy (STS) measurements, and succeed in relating our observation to the properties of SOC. Images acquired on heterostructures with different twist angles between the SLG and WSe$_2$ lattices give fully consistent results as to the type and magnitude of the SOC imprinted by the WSe$_2$ layer onto SLG. We find that both Rashba and valley-Zeeman (vZ) terms are present, with a magnitude varying between 7 and 16 meV for the Rashba contribution, and between 1 and 3 meV for the vZ. These values are in perfect agreement with the results of earlier magnetotransport studies[13–15], and establish that the strength of the Rashba term is one order of magnitude larger than that predicted by existing ab initio calculations[16–19]. We further find that no quasiparticle interference is observed for SLG-on-WSe$_2$ when the twist angle is 30°, in agreement with theoretical expectations[17–20], because for this angle the strength of the vZ coupling vanishes by symmetry and backscattering is only present when both vZ and Rashba are non-zero. Besides settling pending issues about the type and strength of SOC in SLG-on-TMD, our work demonstrates a conceptually new approach to probe spin-orbit coupling quantitatively in 2D materials.

## Spin-orbit coupling in graphene

Given the very weak ($\approx$50 μeV) SOC naturally present in graphene[21], the ability to controllably generate SOC artificially, while preserving the basic electronic properties of Dirac electrons, is of interest in different contexts[16,22–24]. A route towards this goal consists in imprinting SOC by placing graphene on semiconducting TMD substrates such as WSe$_2$, which combine a very large SOC strength ($\approx$500 meV in the valence band)[25,26] with a large band gap around the graphene Fermi level. A variety of transport experiments[13–15,23,27–34] exhibiting unambiguous signatures of a strong SOC in graphene show that this route is successful. However, most experiments detect SOC exclusively through a fast spin relaxation[27–33], much faster than that expected in pristine graphene. This unfortunately provides only limited information, since a fast spin relaxation can be understood in terms of spatially random spin-dependent scattering, and does not require that the SOC imprinted by the TMD substrate corresponds to a modification of the graphene band structure. Therefore, most reported studies which analyse fast spin relaxation times, neither give conclusive evidence for the presence of spin-splitting in the dispersion relation of Dirac electrons, nor do they give unambiguous indications as to the type of SOC imprinted in graphene. Only a few experiments indicate that the SOC imprinted by the TMD substrate onto graphene does induce a modification to the band structure[13–15], and attribute the resulting spin-splitting near the Dirac point to two distinct SOC terms. These are a Rashba term (which wants the electron spin to point in the graphene plane) and a valley-Zeeman term (which forces the electron spin to point in the direction perpendicular to the graphene plane). The experiments consistently suggest a strength of the Rashba and vZ terms of $\approx$15 meV and $\approx$2 meV, respectively (for graphene on WS$_2$). When compared to ab initio calculations[16–19], these results are only partially in agreement: whereas the observed strength of the vZ term indeed corresponds to the expected one, no calculation has so far found a Rashba term as strong as the one extracted from the experimental data. Whether this deviation originates from the assumptions in the theoretical modeling or from problems with the interpretation of the experiments remains to be established. At this stage, it is crucial to push experiments further, by pursuing alternative strategies to probe SOC in graphene on TMDs, capable of providing quantitative information on the strength of the different SOC terms.

# Quasiparticle interference and intravalley backscattering

In the presence of elastic scattering, the wave of an incoming electron with momentum $\vec{k}$ interferes with the scattered wave of momentum $\vec{k}'$ to form a static charge modulation with momentum $\vec{q} = \vec{k}' - \vec{k}$. The real space mapping of these modulations using STS is known as quasiparticle interference (QPI) imaging[35,36]. The Fourier transform (FT) of QPI images gives a finite signal at $\vec{q}$, and interference due to backscattering (i.e. $\vec{k}' = -\vec{k}$) leads to a maximum in the signal at $q_B = 2|\vec{k}|$. This results—if scattering is isotropic—in a characteristic ring in the FT of the spectroscopic images of SLG, a hallmark of intravalley backscattering that can be phenomenologically understood in terms of the phase space available for elastic scattering from $\vec{k}$ to $\vec{k}'$ on a given constant energy contour of the Dirac cone. The phase space (depicted in red in Fig. 1c, d) depends on the scattering momentum $\vec{q}$, and has a distinct maximum when $\vec{k}' = -\vec{k}$ (Fig. 1e), which is why the ring of maximum intensity in Fig. 1g occurs if backscattering is allowed. If backscattering is forbidden, no peak is present (Fig. 1e), and a result of the type shown in Fig. 1f is expected. The presence or absence of peaks at $q_B$ in the QPI thus serves as a direct measure of the existence of backscattering.

We measured the QPI images on different heterostructures consisting of mechanically exfoliated SLG and few-layer WSe$_2$ stacked with selected twist angles between 1° and 30° (Fig. 2a, b—see "Methods"). A typical STM topography of an 18° twist angle heterostructure is shown in Fig. 2c—see Supplementary Section 1 for images of other devices. The twist angle and corresponding moiré pattern can be directly inferred from the Fourier transform of the topography in Fig. 2d. Tunneling spectroscopy (Fig. 2e) reveals the well-known phonon gap at the Fermi level ($E_F$) and the Dirac point, whose shallow depression near −197 mV is consistent with previous studies[37]. Detecting backscattering by QPI relies on large area high resolution $dI/dV(V, \vec{r})$ tunneling conductance maps. Large area scans are necessary to resolve the small momentum components near the Γ-point in Fourier space, which is where intravalley backscattering processes manifest themselves. Figure 3a and b presents such a conductance map and its FT, respectively. Similar to the topography, they reveal the 18° twist angle between the SLG and WSe$_2$ lattice peaks, the corresponding moiré peaks, and their linear combinations. The six ring patterns near the K and K' points correspond to intervalley scattering sketched in Fig. 3c, a well-understood feature previously reported for SLG[38,39].

Figure 3b also shows the presence of *intravalley* scattering contributions around the Γ-point, which are illustrated in full detail in Fig. 4a, with data measured on a 24° twist angle heterostructure. What is unusual for graphene is that the QPI intensity peaks along a ring centered at $k = 0$. Such a behavior has been observed earlier in bilayer graphene[39,40]—where it has been shown to be due to backscattering—but is not expected to occur in pristine SLG, since backscattering is in principle forbidden[38,39]. The observed feature is clearly reminiscent of

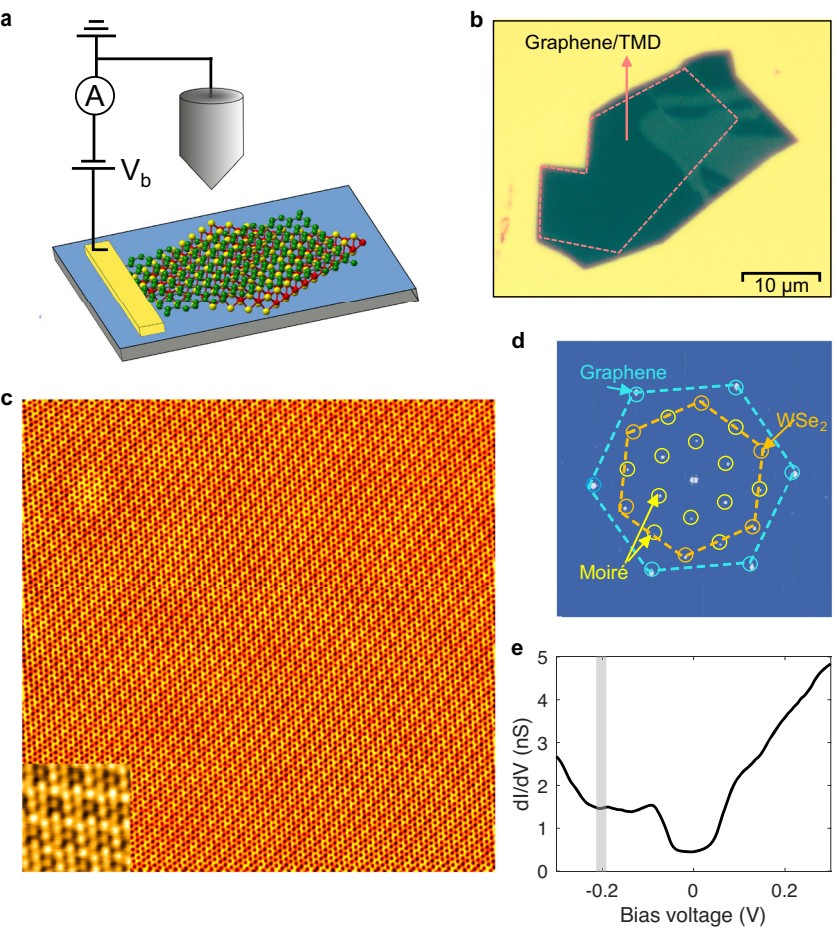

**Fig. 2 | Experimental setup and STM/STS data of an 18° twisted SLG-on-WSe$_2$ heterostructure. a** Schematic view of a twisted SLG on WSe$_2$ assembled onto a SiO$_2$/Si substrate, with gold contact and STM tip. **b** Optical microscopy image of an actual device. **c** 20 × 20 nm$^2$ topographic STM image ($V$ = 20 mV, $I$ = 30 pA; Inset: 2 × 2 nm$^2$ magnification) and **d** corresponding Fourier transform showing the SLG and WSe$_2$ lattices, and the moiré peaks. **e** $dI/dV(V)$ spectrum measured on the bare SLG surface. The gap around the Fermi level is from phonon-assisted inelastic tunneling and the shallow dip near −197 meV (grey bar) is the Dirac point.

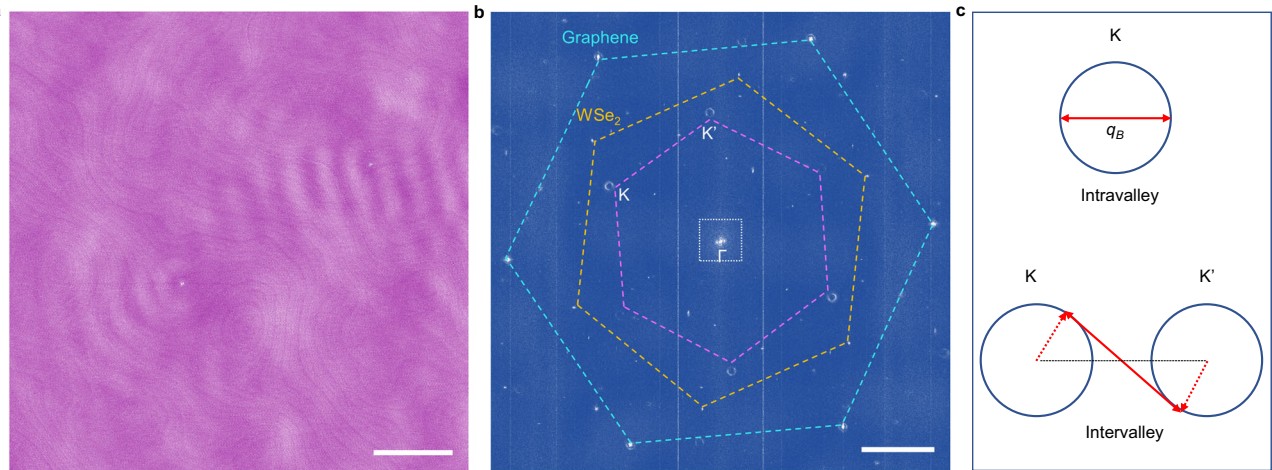

**Fig. 3 | QPI imaging on an 18° twisted SLG-on-WSe₂ heterostructure. a** $dI/dV$ ($V = 30$ mV, $\vec{r}$) map ($I = 50$ pA; scale bar, 30 nm) and **b** corresponding Fourier transform (scale bar, 10 nm⁻¹). Note the finite intravalley scattering amplitude near Γ. **c** Schematics of possible electron scattering wavevectors in graphene.

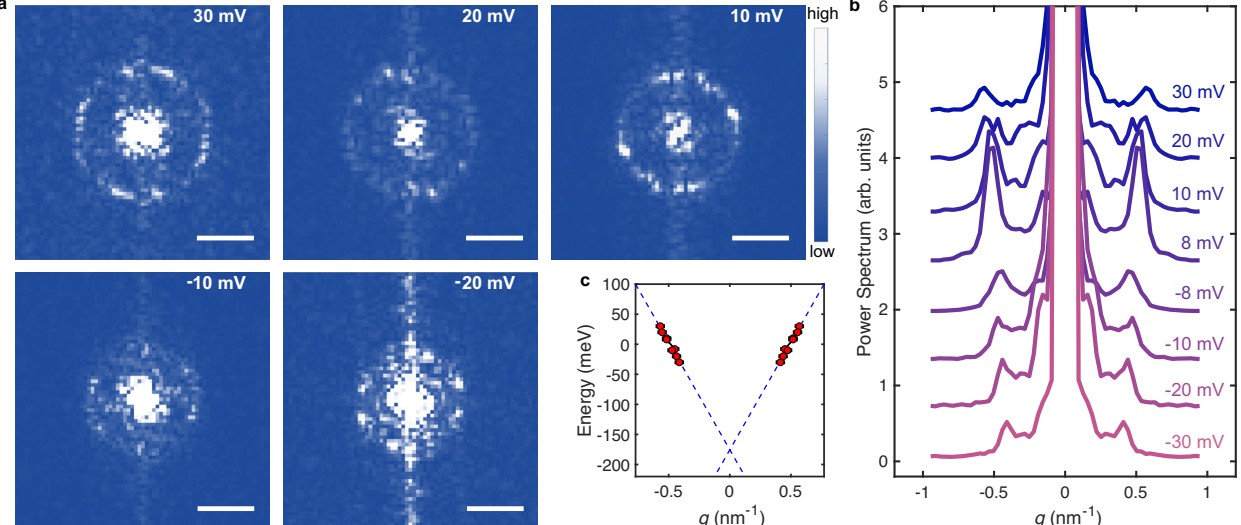

**Fig. 4 | Energy dependent QPI near the Γ point measured on a 24° twisted SLG-on-WSe₂ heterostructure. a** Magnified small $q$ region of the Fourier transforms of different conductance maps $dI/dV(V, \vec{r})$ measured at $V = 30$ mV, 20 mV, 10 mV, −10 mV, and −20 mV over the same area (scale bar, 0.5 nm⁻¹). **b** Angular averaged cuts through the Γ point of the different Fourier transforms. **c** Energy dependence of the backscattering peaks in (**b**), with a linear fit (dashed lines) corresponding to a Fermi velocity $v_F \approx 10^6$ m/s ($E_F = 170$ meV).

the manifestation of intravalley backscattering introduced in Fig. 1g. In that case, the ring is homogeneous (i.e., the intensity is angle-independent), because the model considers isotropic scattering; the inhomogeneous ring observed experimentally is the consequence of the non-uniformity of the scattering potential, i.e. the fact that collisions occur preferentially in some propagation directions, due to the specific spatial distribution of scatterers present in the device.

The evolution of the ring around the Γ-point with electron energy (Fig. 4a) as well as the analysis of the angle-averaged scattering intensity (Fig. 4b) fully confirm that the observed phenomenon is due to backscattering of Dirac electrons. As clearly visible in Fig. 4a, the radius $q_B$ of the ring at which intravalley scattering peaks, decreases with reducing energy from 30 meV to −20 meV. We quantify the precise dependence of $q_B$ on energy by analyzing angle-averaged line cuts (such as the ones shown in Fig. 4b), which allows us to extract the position of the peak in Fig. 4a as a function of applied bias. The resulting electron dispersion (Fig. 4c) is linear as expected for Dirac

electrons in SLG[9], and the Fermi velocity obtained from a linear fit is $v_F \approx 10^6$ m/s (dashed lines in Fig. 4c), in excellent agreement with values obtained for SLG-on-WSe₂ by other techniques[41]. The observed phenomenology is therefore fully consistent with the presence of backscattering (as demonstrated by the ring-shaped peak in intensity around the Γ-point) of Dirac electrons (as demonstrated by the linear dispersion relation with the correct Fermi velocity). We conclude that in contrast to pristine SLG, where no backscattering is observed, backscattering of Dirac electrons is present in SLG-on-WSe₂.

## Quantitative estimate of spin-orbit couplings

Having established the presence of intravalley backscattering of Dirac electrons in SLG-on-WSe₂, we proceed with a fully detailed theoretical analysis of the data. To model this, we add to the usual single-valley Dirac Hamiltonian the Rashba term $H_R = \lambda_R(\tau^z\sigma^x s^y - \sigma^y s^x)$ and the vZ term $H_{vZ} = \lambda_{vZ}\tau^z s^z$, where $\sigma$ refers to the sublattice pseudospin, $\tau$ to the K/K′ valley and $s$ to the physical spin. $H_{vZ}$ acts like a valley-dependent

magnetic field, splitting spin up and down in different directions in the K and K′ valleys, and is a direct descendant of a similar term presents in WSe₂. By symmetry, $\lambda_{vZ}$ must vanish in the heterostructure with a 30° twist angle between graphene and the WSe₂ layer[17–19], which has important experimental consequences (see below). The Rashba term couples spin and pseudospin, leading to spin split bands with opposite pseudospin and spin windings. When both vZ and Rashba are present, the pseudospin vectors are mixed with the spin degree of freedom, and these spin/pseudospin vectors are no longer opposite for opposite momenta, so that nothing prevents backscattering. Importantly—since *both* the vZ and Rashba coupling
s have to be non-zero for the argument to hold—backscattering is expected to disappear for a 30° twist angle between SLG and WSe₂, when symmetry imposes that $\lambda_{vZ} = 0$.

Using the model Hamiltonian with the vZ and Rashba terms, we have calculated the theoretical QPI patterns (see Supplementary Section 2 for details of the calculations) for bare SLG, and for SLG-on-WSe₂ with SOC, in the limit of small disorder. We have also modelled the effect of a mass gap, by adding to the Hamiltonian a spin-independent term of the type $H_\Delta = m\sigma^z$, because such a term can also induce backscattering in SLG. However, in contrast to SOC, whose presence for SLG-on-TMD is established, there is no earlier experimental indication that a sizable mass gap is present in these heterostructures: a sizable mass gap should lead to an insulating behavior at low-temperature, which has never been reported in transport experiments (in contrast to the analogous case of well-aligned SLG-on-hBN, where an insulating behavior due to a mass gap of 20 meV has been observed repeatedly[42,43]).

The results of the calculations for the different scenarios are illustrated in Fig. 5b–d, and by the continuous line curves in the top panels of Fig. 5f–h. As expected, there is no ring at $q_B$ in the FT of the QPI pattern for bare SLG in Fig. 5b, whereas a clear backscattering ring is present when either SOC (Fig. 5c) or a mass gap (Fig. 5d) are included. To discriminate which one of the two cases captures the physics of SLG-on-WSe₂, we analyze the theoretically predicted angular-averaged profiles of the QPI patterns, shown by the continuous line in the top panels of Fig. 5g and h for SOC and for a mass gap, respectively. Their comparison shows two clear differences. First, the mass gap causes a single very large peak with an extremely pronounced asymmetry, whereas SOC leads to three peaks, resulting in a structure that is considerably more symmetric around the central one. Second, the mass gap gives a tail in the profile that, at large $q$, decays much more slowly than that calculated for SOC.

To *quantitatively* compare the experimental results with the theoretically predicted profiles, we broaden the calculated curves (dashed lines in Fig. 5f–h) to account for the finite experimental resolution. This is due to the momentum resolution brought upon by the finite image size, and electrostatic potential fluctuations in graphene (i.e., an electron interfering at constant energy changes the magnitude of its wavevector during propagation, an effect that is not taken into account by the Hamiltonian that we use to describe the system, see Supplementary Section 2). To maximize the signal, the comparison is done with QPI profiles obtained from an angular average of the experimental QPI pattern in Fig. 5e–h taken over a finite angle in Fig. 5a. This also allows us to eliminate parasitic experimental effects, such as the non-zero signal at $q_x = 0$ in Fig. 5a, which is an artefact originating from the STM tip scanning. We have checked that the analysis of different angular sectors with sufficiently large signal leads to identical results (i.e., our conclusions are robust—see Supplementary Section 3).

23 QPI images measured on four devices with different twist angles between 1° and 30° were analyzed in detail. The data unambiguously show that the best agreement with calculations is obtained

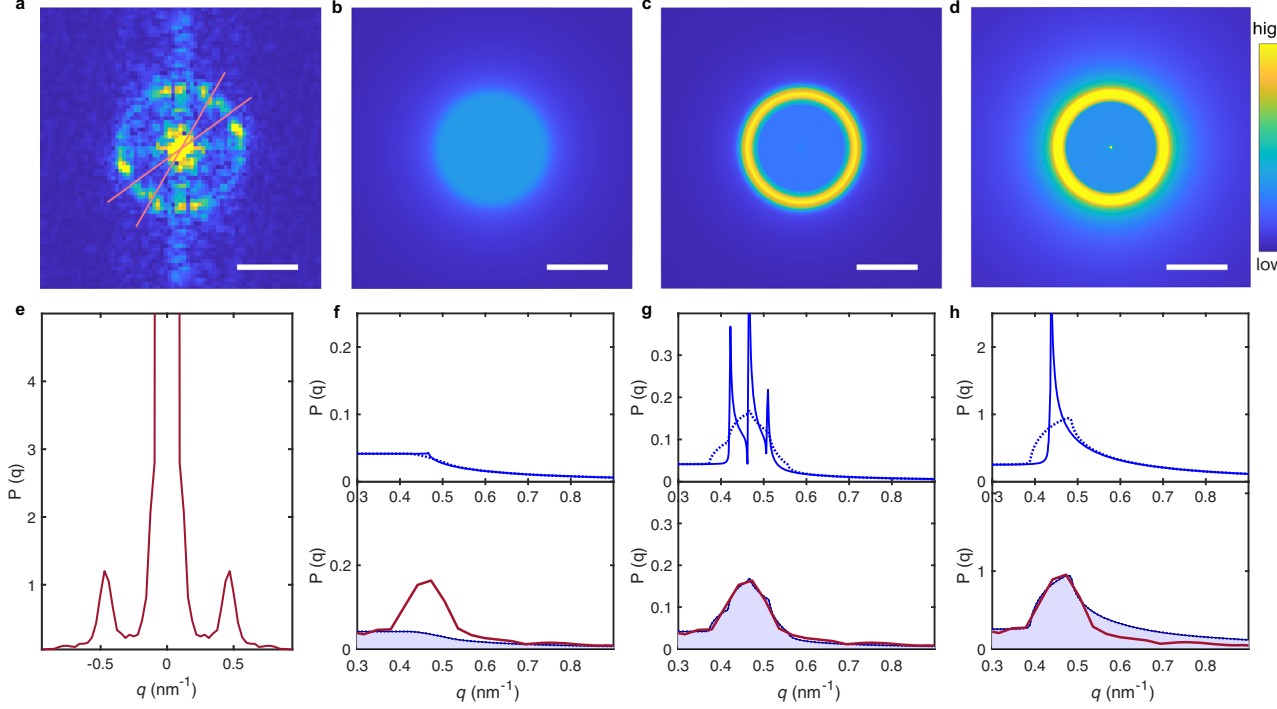

**Fig. 5 | Theoretical modelling of the QPI patterns around the Γ point. a** Small $q$ Fourier transform of a conductance map measured at 10 mV on a 24° twisted SLG-on-WSe₂ heterostructure. **b–d** Corresponding QPI patterns calculated for bare SLG, SLG with SOC, and SLG with a mass gap, respectively (scale bar, 0.5 nm⁻¹). **e** Angular averaged line cut through the experimental data in (**a**), revealing distinct backscattering peaks at ± 0.47 nm⁻¹. **f–h** The solid blue lines are the calculated QPI spectra for the three cases we considered: **f** bare SLG; **g** SLG with SOC; **h** SLG with a mass gap. The dotted blue lines represent the theoretical curves broadened to account for momentum resolution and potential fluctuations. Their purple-shaded integral is fanned out over $2\pi$ to construct the model QPI in panels **b–d**. They are compared to the experimental red scattering profile at the bottom of panels (**f–h**), with SOC matching the data best. Model parameters are: in **b**, **f**: $\lambda_{vZ} = 0$ mV, $\lambda_R = 0$ mV, $m = 0$ mV; in **c**, **g**: $\lambda_{vZ} = 2.0$ mV, $\lambda_R = 15$ mV, $m = 0$ mV; and in **d**, **h**: $\lambda_{vZ} = 0$ mV, $\lambda_R = 0$ mV, $m = 12$ mV.

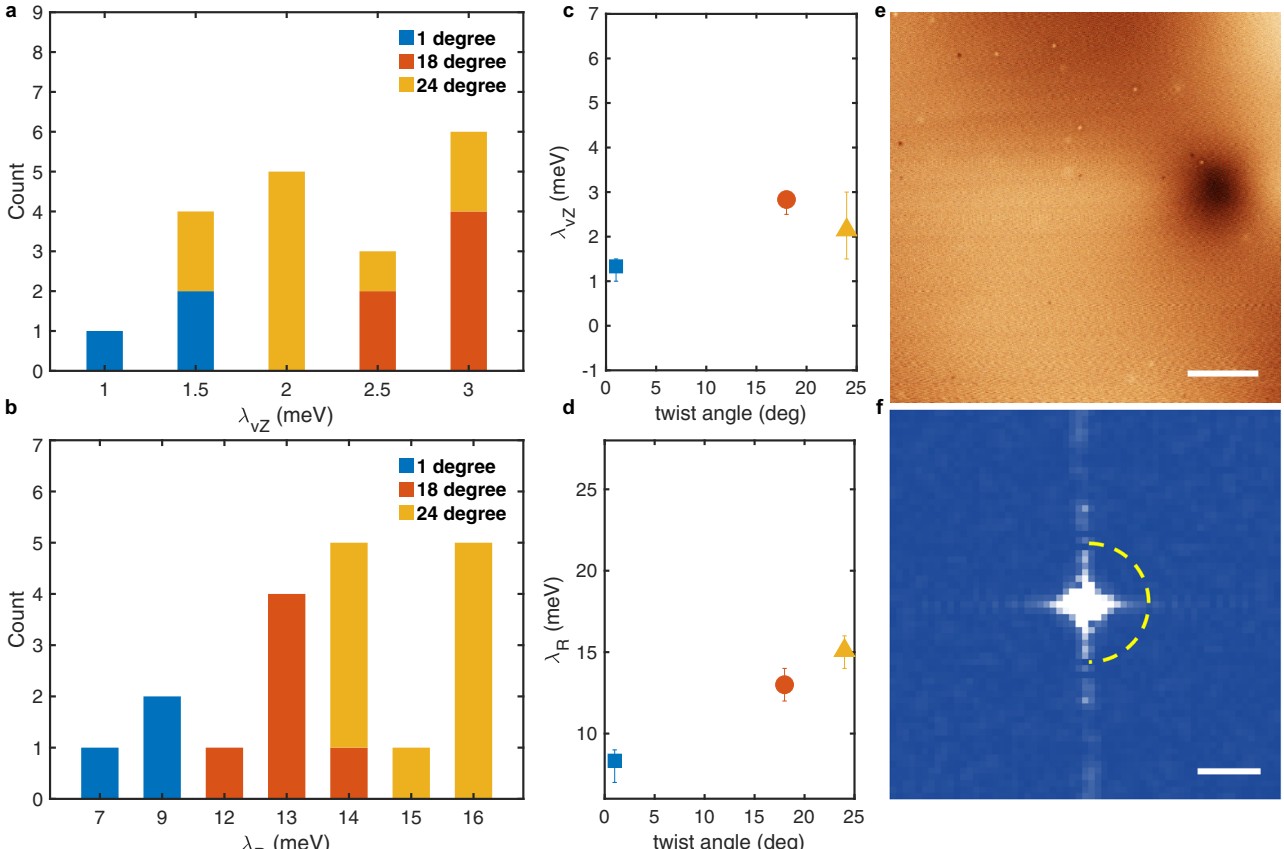

**Fig. 6 | Summary of the SOC model fitting of all the devices we measured.**
**a**, **b** Histograms of $\lambda_{vZ}$ and $\lambda_R$ extracted from 19 QPI patterns measured on the
devices with twist angles from 1° to 24° based on the SOC model. **c**, **d** Proximity
induced $\lambda_{vZ}$ and $\lambda_R$ SOC terms in SLG as a function of twist angles. Each symbol
corresponds to the average value obtained for a given twist angle, with error bars
reflecting the SOC range in the fitting. **e** STM Topography of a 30° twist angle
heterostructure (scale bar, 20 nm) showing several defects. **f** FFT of a $dI/dV$
($V = 50$ mV) map acquired in the same area as (**e**) (scale bar, 0.5 nm⁻¹). In the pre-
sence of intravalley backscattering, one would expect a ring at the position indi-
cated by the yellow dashed arc. None of the four $dI/dV(V, \vec{r})$ maps acquired on the
30° device show the presence of intravalley backscattering.

for the scenario based on backscattering originating from SOC. The
result is illustrated by comparing theory and data measured on the 24°
twist angle device. Figure 5g,h show the measured data (red line) and
calculated curves (blue line, delimiting the purple-shaded region)
including either SOC or a mass gap. The curve calculated with SOC
overlays perfectly with the data over the entire $q$-range investigated,
for a strength of the vZ and of the Rashba terms of $\lambda_{vZ} = 2.0$ meV and
$\lambda_R = 15$ meV. In other devices, comparable agreement is found with $\lambda_{vZ}$
ranging between 1–3 meV and $\lambda_R$ between 7–16 meV, as summarized in
Fig. 6a, b. These values coincide nearly perfectly with estimates based
on earlier transport experiments[13–15]. In contrast, for a finite mass gap
(Fig. 5h) the agreement with data is poorer, as theory predicts a much
slower intensity decay at large $q$ than what is observed in the experi-
ments. Additionally, the curve exhibiting the best agreement corre-
sponds to $m = 12$ meV, or equivalently to a gap of 24 meV. This value
should have led to a detectable insulating behavior in at least some of
the earlier transport experiments, but in no case such an insulating
behavior has been reported. Hence, a value of $m = 12$ meV not only
gives poorer agreement with the data than the SOC scenario, but it also
is incompatible with existing transport experiments, and can therefore
be discarded.

These results demonstrate that the analysis of QPI images in SLG-
on-WSe₂ allows establishing the presence of SOC, and to determine the
type and strength of the different contributions. On this basis, we
conclude that the strength of the Rashba term $\lambda_R$ is indeed an order of
magnitude larger than ab initio predictions[16–19], consistent with earlier
transport experiments. Interestingly, the data measured on our

devices seem to show some systematic dependence in the strength of
the two SOC terms on twist angle (even though a larger statistics would
be desirable to establish such a conclusion firmly). For the vZ term, the
strength is approximately constant upon varying twist angle (Fig. 6c);
for the Rashba term, the strength appears to increase with increasing
angle between 1° and 24° (Fig. 6d). The error bars correspond to the
range of all the valley-Zeeman and Rashba terms obtained for each
twist angle. Finding experimental indications of a systematic angular
dependence of the SOC strength, motivated us to verify whether a
device with 30° twist angle does show no sign of backscattering, as
expected theoretically (since $\lambda_{vZ} = 0$ in that case). The result of the
experiments is shown in Fig. 6e, f, and is precisely what is expected in
the absence of backscattering: no QPI is observed neither in real space
(Fig. 6e) nor in the FT (Fig. 6f; see also Supplementary Figure S4),
despite the fact that the topographic images on the 30° device appear
to be of identical quality as those on devices with other twist angle (see
Supplementary information S1 and S4). Such an observation strongly
confirms the validity of our approach to measure the strength of SOC
in SLG-on-WSe₂ and validates our conclusions as to the strength of the
vZ and of the Rashba terms.

Determining what type of SOC is present in the band structure of
an electronic system and estimating the magnitude quantitatively are
notoriously complex issues. On 2D materials, only a few techniques
have been employed so far to extract this information, all based on the
study of low-temperature magnetotransport as a function of gate
voltage in devices with sufficient high mobility. The use of scanning
tunneling spectroscopy to perform quasiparticle interference imaging

to achieve this same goal demonstrates a strategy that had not been pursued earlier. It relies on a specific phenomenon—in the present case, whether backscattering occurs or not—that can be probed by careful STM measurements and modeled in detail theoretically. As we have shown here, on SLG-on-WSe$_2$ this technique allows a full characterization of the SOC terms present, which is key to establish a firm experimental understanding of the spin-orbit coupling imprinted in graphene by proximity effect. We anticipate that the analysis of similar QPI imaging measurements on other material systems, in conjunction with detailed theoretical modeling, will also enable to obtain detailed information about spin-orbit coupling.

## Methods

### Assembly of the SLG-on-WSe$_2$ heterostructures

The devices were fabricated using the dry transfer method[44]. An exfoliated single-layer graphene and a few-layer thin WSe$_2$ crystal were picked up with a defined twist angle by a propylene carbonate (PC) film coated on polydimethylsiloxane (PDMS) stamps. This stack was subsequently released on a 90 nm SiO$_2$/Si substrate (Fig. 2c). Gold contacts were applied to the assembled devices using electron-beam lithography.

### STM experiments

All STM data were acquired in ultra-high vacuum (base pressure $< 10^{-10}$ mBar at room temperature) at 4.5 Kelvin using an electrochemically etched tungsten tip. The tips were further conditioned in situ on Au(111). The bias voltage is applied to the sample, with 0 V corresponding to the Fermi level. To acquire reproducible high-resolution scanning tunneling microscope (STM) images and $dI/dV(V, \vec{r})$ scanning tunneling spectroscopy (STS) maps, each device was thoroughly cleaned in three steps. First, lithographic residues were removed by scanning the graphene surface with an atomic force microscope tip in air[13]. Second, the devices were annealed at 350 °C for 180 min in a H$_2$/Ar atmosphere before transferring them into the STM chamber. Third, the devices were annealed once more at 350 °C for an hour in ultra-high vacuum. Topographic STM images were acquired in constant-current mode. The $dI/dV(V, \vec{r})$ maps were measured using a standard lock-in technique with a bias modulation amplitude $V_{rms} = 2$ mV at a frequency $f = 429$ Hz. The $dI/dV(V, \vec{r})$ map in Fig. 3a has 1848 × 1848 pixels and was acquired with a scanning speed of 21 nm/s. The other $dI/dV(V, \vec{r})$ maps have 256 × 256 pixels and were acquired at a scanning speed of 20 nm/s.

## Data availability

Data associated with this study are freely available on Yareta (https://doi.org/10.26037/yareta:p45pwzleczgidm4buoiuuqxw3e).

## Code availability

Codes to analyse the data and perform numerical calculations are available upon reasonable request.

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

## Acknowledgements
We thank A. Guipet for his technical assistance with the scanning probe instrumentation and A. Ferreira for technical help with the instrumentation used for the realization of the devices employed in this study. We thank V. Multian for aligning WSe$_2$ for the 30° twist angle device. This work was supported by the Swiss National Science Foundation (Division II Grant No. 182652). A.F.M. gratefully acknowledges support from the SNF Division II and from the EU Graphene Flagship project. L.R. was funded by the SNSF via Ambizione grant PZOOP2_174208 and via Starting Grant TMSGI2_211296. Z.W. acknowledges support from the National Natural Science Foundation of China (Grants No. 11904276).

## Author contributions
C.R. and A.F.M. proposed and designed the experiment. D.M., L.S., I.G.L., and Z.W. participated in preparing the devices. L.S. performed the STM measurements with the help from A.S., Á.P., and J.M.C. L.R. carried out the calculations and provided theoretical support. L.S., L.R., A.S., Á.P., A.F.M., and C.R. discussed and analyzed the data. L.S., L.R., A.F.M., and C.R. wrote the manuscript. All authors read and commented on the manuscript.

## Competing interests
The authors declare no competing interests.
