## [Peer Review File · Nature Communications]

Reviewers' Comments:

Reviewer #1:

Remarks to the Author:

Referee report of manuscript NCOMMS-23-00424-T by Lihuan Sun et al.

The authors study the induced spin-orbit coupling (SOC) in monolayer graphene placed on top of a transition metal dichalcogenide flake. Using scanning tunneling microscope they can image quasiparticle interference patterns. The analysis of the experimental results helped to determine both the type and strength of the induced SOC terms. This methodology allows a more direct determination of the induced SOC than earlier methods, which were based e.g., weak localization transport measurements. The magnitude of the valley Zeeman type SOC agrees quite well with earlier theoretical predictions and experimental results. In contrast, the strength of Rashba type SOC extracted from the presented experiments appears to be significantly larger than theoretical predictions.

In general, proximity induced properties can be very important in stacks of atomically thin materials. This work is a significant contribution in this respect. It is also clearly written and readable. The apparently much stronger Rashba type SOC than found in earlier works should motivate further theoretical and experimental investigations.

I can recommend the publication of this manuscript once the authors considered the following remarks/questions.

Remarks/Questions

1) The authors write in the caption of Fig.1:

" b) Pseudospin and spin textures in the presence of SOC".

I find this sentence misleading and difficult to understand. The figure shows what one can call a pseudospin winding and the spin-orbit split bands of graphene. However, there is no information on the spin texture: for the measured strong Rashba type SOC the spin expectation value would be mostly in-plane and tangential to the Fermi surface at all k-points. In contrast, the arrows in Fig1.b are perpendicular to the Fermi surface. Therefore they are not an illustration of the spin texture (except for special interlayer twist angles, as discussed recently in Ref.20 of the manuscript)

2) The picture shown in e.g. Fig.3 (a) suggests that parameters of the induced SOC were determined from information that corresponds to a roughly 200nm x 200nm area of the sample. In transport experiments the channel length and width is typically larger than this and in order to be able to compare the results of this work to e.g. weak localization experiments one would need to know if the induced SOC is homogeneous throughout the sample.

Have the authors tried to determine the SOC parameters using the same graphene/WSe2 samples but in different areas on the sample?

Reviewer #2:

Remarks to the Author:

The current manuscript deals with interesting experimental results of QPI on graphene/WSe2 heterostructures. The experiments are well-designed. The claims made are supported by the data. The manuscript is well-written.

However, it is not clear what is the main takeaway from this work. There are two aspects: (1) the presence of back-scattering in graphene on WSe2 when the relevant spin and pseudospin degrees of freedom are mixed and (2) the estimation of the various components of SOC. The demonstration of the first point is interesting and validates theoretical expectations. On the other hand, the second aspect has been dealt with extensively via transport measurements. The authors might consider highlighting the first aspect more in their title and abstract (based solely on the

second aspect, the manuscript would not have been suitable for publication in Nat Comm.).

Before the manuscript can be recommended for publication, I would like to get clarification on the following:

1. The authors claim (line 125) that most studies do not establish spin-splitting of the bands. This is not entirely true - studies have shown the spin-splitting from measurements of SdH oscillations in graphene on WSe₂.
2. Similarly, the authors' claim that the type of SOC is not well settled (lines 128-129) is also not entirely accurate. Low-temperature transport measurements provide the magnitudes of both Rashba and VZ SOC. I agree that these estimates can vary depending on the model one uses to analyze the data (but so are the conclusions of this manuscript). The authors might consider including this point in their introductory discussion.
3. Why the dip at the Dirac point (Fig. 2(e)) is so shallow?
4. The authors should present a comparison of the QPI data on single-layer graphene and graphene/WSe₂ heterostructure.
5. A minor point: Other than the points marked in the FT plot of Fig 3(b), one can see several points of comparable intensity. The authors should comment on their origin.

Reviewer #3:

Remarks to the Author:

The authors present quasiparticle interference (QPI) studies on single-layer graphene/WSe₂ heterostructures with different twist angles. The authors observed an intravalley backscattering ring in the QPI patterns for their graphene/WSe₂ samples with 1-degree, 18-degree, and 24-degree twist angles. The authors attributed this observation to WSe₂-induced spin-orbit coupling (SOC) in graphene bands. To verify this interpretation, the authors further did QPI measurements on a graphene/WSe₂ sample with a 30-degree twist angle, which is theoretically predicted to have vanishing intravalley backscattering ring according to the WSe₂-induced SOC interpretation. And the authors confirmed the disappearance of QPI rings in such a scenario with their experiment. Furthermore, the authors compared the angle averaged QPI power spectra between the experiment and theory to extract the detailed quantities of WSe₂-induced SOC in graphene, including the strength of valley-Zeeman SOC and Rashba SOC. And the authors found agreement between their extracted values with those extracted from previous transport experiments but disagreement with those from DFT calculations.

Inducing and controlling SOC in 2D materials is a timely study with broad appeal. In this work, the authors demonstrated the possibility of using QPI to extract detailed quantitative SOC parameters for substrate-induced SOC in graphene and further demonstrated the control of SOC by controlling the twist angle between graphene and the WSe₂ substrate. Overall, the experimental data and discussions are clear in this manuscript, so I recommend the publication of this manuscript in Nature Communications. But there are several issues I would like the authors to address:

- 1) Could the authors add a more detailed fitting procedure for how the valley-Zeeman and Rashba SOC strength are extracted from the experimental QPI power spectra? And how the extraction errors are assigned?
- 2) Could the authors comment on how many different regions within each device the authors have studied? And do they always show similar QPI patterns within the same device with separately calibrated STM tips?
- 3) For the 30-degree twist angle sample, does the QPI ring always vanish for various scanning bias voltages?

4) Could the authors add the pixel number and scanning speed information for the dI/dV maps used for the QPI study?

5) If the authors have also done QPI measurements on small twist angle graphene/hBN samples, it would be nice for the authors to add such data to the manuscript to demonstrate that, in this case, the QPI power spectra follow the theoretically calculated QPI power spectra for graphene with a mass gap as shown in Fig. 5h.

Response to reviewers of “Determining spin-orbit coupling in graphene by quasiparticle interference imaging”.

Report of Referee#1 – NCOMMS-23-00424-T

The authors study the induced spin-orbit coupling (SOC) in monolayer graphene placed on top of a transition metal dichalcogenide flake. Using scanning tunneling microscope they can image quasiparticle interference patterns. The analysis of the experimental results helped to determine both the type and strength of the induced SOC terms. This methodology allows a more direct determination of the induced SOC than earlier methods, which were based e.g., weak localization transport measurements. The magnitude of the valley Zeeman type SOC agrees quite well with earlier theoretical predictions and experimental results. In contrast, the strength of Rashba type SOC extracted from the presented experiments appears to be significantly larger than theoretical predictions.

Reply: We thank Referee#1 for their summary of our work. Here, we would like to emphasize an important aspect of our study. Quasiparticle interference (QPI) imaging is not the only technique capable of identifying the nature of the SOC terms and determining their strength experimentally. We cite three previous studies in our manuscript (Wang *et al.*, 2016; Island *et al.*, 2019; Wang *et al.*, 2019) which conclude that there is a band structure modification and which find a valley Zeeman (vZ) and a Rashba SOC term. The vZ strength found in these studies is in agreement with theoretical calculations. However, the Rashba term they find is an order of magnitude larger than what theory predicts. This discrepancy is at the origin of an ongoing debate: since the model calculations correctly predict the vZ term, it is commonly anticipated that they also give the correct value for the Rashba term, which would thus imply that experiments are wrong. The importance of our QPI study in this context is that we find the same amplitudes for the vZ and Rashba terms as transport experiments, but in a completely different and independent experiment. Finding such a consistent experimental amplitude for the Rashba term on similar devices using completely different experiments is a strong indication that it is correct, and that some ingredient is missing in the theoretical calculations. Finally, we stress that both our QPI pattern and the Shubnikov-de Haas oscillations imply a modified band structure. This is not necessarily the case for weak antilocalization, which can be understood solely in terms of impurity scattering, without involving any modification of the pseudospin texture.

In general, proximity induced properties can be very important in stacks of atomically thin materials. This work is a significant contribution in this respect. It is also clearly written and readable. The apparently much stronger Rashba type SOC than found in earlier works should motivate further theoretical and experimental investigations.

Reply: As already mentioned, there are several earlier experiments which also found a much larger Rashba term, in excellent agreement with what we find by analyzing QPI patterns. We cite three of them in our manuscript. The discrepancy we refer to is with respect to model calculations, which find a much smaller Rashba term. The importance of our study is to propose another,

completely independent experiment to determine the amplitude of the Rashba term. The quantitative agreement of our data with earlier experiments which also find a large Rashba term strongly indicates that theory is missing something when calculating the Rashba contribution.

I can recommend the publication of this manuscript once the authors considered the following remarks/questions.

Reply: We thank the referee for their positive comments on our manuscript and for recommending its publication. In the following, we respond in details to their comments.

Comments/Questions:

1. The authors write in the caption of Fig.1: “b) Pseudospin and spin textures in the presence of SOC”. I find this sentence misleading and difficult to understand. The figure shows what one can call a pseudospin winding and the spin-orbit split bands of graphene. However, there is no information on the spin texture: for the measured strong Rashba type SOC the spin expectation value would be mostly in-plane and tangential to the Fermi surface at all k-points. In contrast, the arrows in Fig1.b are perpendicular to the Fermi surface. Therefore they are not an illustration of the spin texture (except for special interlayer twist angles, as discussed recently in Ref.20 of the manuscript)

Reply: We thank Referee#1 for pointing out this mistake. The key ingredient to understand our QPI images is the pseudospin texture. When it is purely radial, backscattering is not possible because the k and $-k$ pseudospins are antiparallel. Backscattering only becomes possible when the pseudospin is canted out of plane. This is what we want to illustrate with Fig. 1 - there is no spin texture in that representation of the Dirac cones. We have corrected the caption of Fig. 1b accordingly in the revised manuscript as shown below:

Response Figure 1: **Pseudospin winding in graphene and QPI patterns with and without backscattering.** **a**, Pseudospin winding (black arrows) in a single Dirac cone. **b**, Pseudospin winding and spin-orbit split bands of graphene in the presence of SOC. Blue and red arrows illustrate the tilted pseudospin winding. The blue (red) Dirac cone corresponds to spin up (down) configuration.

2. The picture shown in e.g. Fig.3 (a) suggests that parameters of the induced SOC were determined from information that corresponds to a roughly 200nm x 200nm area of the sample. In transport experiments the channel length and width is typically larger than this and in order to be able to compare the results of this work to e.g. weak localization experiments one would need to know if the induced SOC is homogeneous throughout the sample. Have the authors tried to determine the SOC parameters using the same graphene/WSe₂ samples but in different areas on the sample?

Reply: We have measured different locations on different samples. The SOC terms and the twist angle are reproducible and homogeneous in each of the devices we have investigated. We have added the analysis of the QPI pattern in four additional regions from the original data sets to figure Fig. 6. The strengths of the SOC terms extracted from the QPI analysis at different energies and positions for a given twist angle vary by about 10%. The twist angle in a given device varies by at approximately 1°.

Report of Referee#2 – NCOMMS-23-00424-T

The current manuscript deals with interesting experimental results of QPI on graphene/WSe₂ heterostructures. The experiments are well-designed. The claims made are supported by the data. The manuscript is well-written.

Reply: We thank Referee#2 for the positive comments about our manuscript, and for pointing out that our data are convincing.

However, it is not clear what is the main takeaway from this work. There are two aspects: (1) the presence of back-scattering in graphene on WSe₂ when the relevant spin and pseudospin degrees of freedom are mixed and (2) the estimation of the various components of SOC. The demonstration of the first point is interesting and validates theoretical expectations. On the other hand, the second aspect has been dealt with extensively via transport measurements. The authors might consider highlighting the first aspect more in their title and abstract (based solely on the second aspect, the manuscript would not have been suitable for publication in Nat Comm.).

Reply: Our manuscript contains indeed the two key messages pointed out by Referee#2. 1) We find that the presence of SOC is directly related to a finite backscattering amplitude, which we detect in QPI images. In the absence of SOC, backscattering is forbidden and the corresponding QPI pattern is absent. 2) A detailed analysis of the QPI pattern allows us to determine the nature of the SOC and to extract the amplitude of the SOC terms in single layer graphene (SLG) on WSe₂. We are fully aware that this information is not unique to our experiment, as pointed out by the referee. But this is precisely the added value of our work, with a totally independent contribution to the ongoing debate in the community as to whether the strength of the Rashba term is correct, because ab-initio calculations predict a 10 times smaller magnitude.

As already mentioned in our response to Referee#1, the importance of our manuscript is to find the same SOC imprinted in SLG by WSe₂ as earlier experiments, but with a completely independent technique. Finding the same v_Z and Rashba amplitudes from QPI and from transport measurements gives a strong confidence in the experimental value. Hence, the disagreement of the Rashba amplitude with theory indicates a missing ingredient in the modeling, rather than a problem of the experiments. Therefore, the correlation between backscattering and the presence of SOC, and the estimation of the SOC parameters are two equally important results of our work.

Before the manuscript can be recommended for publication, I would like to get clarification on the following:

Comments/Questions:

1. The authors claim (line 125) that most studies do not establish spin-splitting of the bands. This is not entirely true - studies have shown the spin-splitting from measurements of SdH oscillations in graphene on WSe₂.

Reply: Referee#2 is right in pointing out that previous studies have provided evidence of the spin-splitting of the bands. We actually discuss these experiments (Wang *et al.*, 2016; Island *et al.*, 2019; Wang *et al.*, 2019) in the manuscript (lines 130-134):

Only a few experiments indicate that the SOC imprinted by the TMD substrate onto graphene does induce a modification to the band structure, and attribute the resulting spin-splitting near the Dirac point to two distinct SOC terms.

One of these publications includes two coauthors of the present manuscript (Wang *et al.*, 2016).

The statement on lines 125-129 pointed out by Referee#2 is referring to experiments which detect SOC via the analysis of fast spin relaxations. The latter do not unambiguously establish spin-splitting of the bands, since they imply spin relaxations which can be understood solely in terms of impurity scattering. To make this point more clear, we have modified the statement on lines 125-129 in the revised manuscript, replacing:

~~Therefore, most reported studies neither give conclusive evidence for the presence of spin-splitting in the dispersion relation of Dirac electrons, nor do they give unambiguous indications as to the type of SOC imprinted in graphene.~~

with

Therefore, most reported studies which analyse fast spin relaxation times, neither give conclusive evidence for the presence of spin-splitting in the dispersion relation of Dirac electrons, nor do they give unambiguous indications as to the type of SOC imprinted in graphene.

2. Similarly, the authors' claim that the type of SOC is not well settled (lines 128-129) is also not entirely accurate. Low-temperature transport measurements provide the magnitudes of both Rashba and VZ SOC. I agree that these estimates can vary depending on the model one uses to analyze the data (but so are the conclusions of this manuscript). The authors might consider including this point in their introductory discussion.

Reply: This question brings up the second key message of our manuscript as already discussed above. There are many transport measurements (eg: weak antilocalization or spin-valve study) which do prove strong SOC on graphene/WSe₂ heterostructure but are not enough for identifying the microscopic mechanism responsible for SOC. A few low-temperature transport measurements provide the magnitudes of both Rashba and vZ SOC, however the strength of the Rashba term found in these experiments is an order of magnitude larger than what theory predicts. The importance of our QPI study in this context is that we find the same amplitudes for the vZ and Rashba terms as transport experiments, but with a completely different and independent experiment. This strongly suggests that experiments are correct, and that some key ingredient is missing in the theoretical description of the system.

3. Why the dip at the Dirac point (Fig. 2(e)) is so shallow?

Reply: The vast majority of tunneling spectra measured by STM on graphene show a weak dip at the Dirac point, very similar to what we observe in Fig. 2(e). This dip is occasionally a little deeper as in Supplementary Fig. S4d. In all STM experiments, it is identified as the Dirac point because it is shifting as expected when changing the doping with a gate voltage. Why this dip is mostly very shallow is not understood at this time.

4. The authors should present a comparison of the QPI data on single-layer graphene and graphene/WSe₂ heterostructure.

Reply: Our manuscript is focusing on the appearance of backscattering in the presence of SOC, and on the twist angle dependence of the corresponding QPI features in graphene on WSe₂. QPI imaging of single layer graphene without SOC has been extensively discussed and is the subject of numerous previous publications. They all agree that without SOC, there is no QPI feature corresponding to backscattering -we cite two representative references in our manuscript.

5. A minor point: Other than the points marked in the FT plot of Fig 3(b), one can see several points of comparable intensity. The authors should comment on their origin.

Reply: Apart from the points we marked in Fig 3(b), the other points of comparable intensity correspond to the moiré pattern and to their linear combination with the Bragg peaks (see simulated image in Supplementary Fig. S1, which also reveals additional peaks). We amend the text to clarify this point by replacing

~~Similar to the topography, they reveal the 18° twist angle between the SLG and WSe₂ lattice peaks and the corresponding moiré peaks.~~

by

Similar to the topography, they reveal the 18° twist angle between the SLG and WSe₂ lattice peaks, the corresponding moiré peaks, and their linear combinations.

Report of Referee#3 – NCOMMS-23-00424-T

The authors present quasiparticle interference (QPI) studies on single-layer graphene/WSe₂ heterostructures with different twist angles. The authors observed an intravalley backscattering ring in the QPI patterns for their graphene/WSe₂ samples with 1-degree, 18-degree, and 24-degree twist angles. The authors attributed this observation to WSe₂-induced spin-orbit coupling (SOC) in graphene bands. To verify this interpretation, the authors further did QPI measurements on a graphene/WSe₂ sample with a 30-degree twist angle, which is theoretically predicted to have vanishing intravalley backscattering ring according to the WSe₂-induced SOC interpretation. And the authors confirmed the disappearance of QPI rings in such a scenario with their experiment. Furthermore, the authors compared the angle averaged QPI power spectra between the experiment and theory to extract the detailed quantities of WSe₂-induced SOC in graphene, including the strength of valley-Zeeman SOC and Rashba SOC. And the authors found agreement between their extracted values with those extracted from previous transport experiments but disagreement with those from DFT calculations.

Inducing and controlling SOC in 2D materials is a timely study with broad appeal. In this work, the authors demonstrated the possibility of using QPI to extract detailed quantitative SOC parameters for substrate-induced SOC in graphene and further demonstrated the control of SOC by controlling the twist angle between graphene and the WSe₂ substrate. Overall, the experimental data and discussions are clear in this manuscript, so I recommend the publication of this manuscript in Nature Communications.

Reply: We thank Referee#3 for the nice summary of our work and for recommending its publication in Nature Communications.

But there are several issues I would like the authors to address:

Comments/Questions:

1. Could the authors add a more detailed fitting procedure for how the valley-Zeeman and Rashba SOC strength are extracted from the experimental QPI power spectra? And how the extraction errors are assigned?

Reply: We have added the following information about the fitting procedure in Section S2.D of the Supplementary Material:

The valley-Zeeman and Rashba SOC are extracted by adjusting the theoretical model to the QPI profile in momentum space. The calculated QPI depends on the following parameters: the valley-Zeeman and the Rashba SOC terms, the sample bias, the Fermi velocity, and the finite momentum resolution of the experimental data given to first order by the image size. Since there are two different kinds of disorder, we use the sum of Λ_0 and Λ_3 power spectra to fit the experimental results. The free parameters are the valley-Zeeman and the Rashba SOC amplitudes. The most suitable valley-Zeeman and Rashba terms are obtained by adjusting the calculated backscattering amplitude to the experimental QPI profile. The width of the backscattering ring is primarily given by the Rashba term and its amplitude by the vZ term.

The error bars in Fig. 6c and Fig. 6d correspond to the spread of the extracted valley-Zeeman and Rashba terms in each device. To make this more clear, we modified the revised manuscript by replacing

~~For the vZ term, the strength is approximately constant upon varying twist angle (Fig. 6c); for the Rashba term, the strength appears to increase with increasing angle between 1° and 24° (Fig. 6d).~~

with

For the vZ term, the strength is approximately constant upon varying twist angle (Fig. 6c); for the Rashba term, the strength appears to increase with increasing angle between 1° and 24° (Fig. 6d). The error bars correspond to the range of all the valley-Zeeman and Rashba terms obtained for each twist angle.

2. Could the authors comment on how many different regions within each device the authors have studied? And do they always show similar QPI patterns within the same device with separately calibrated STM tips?

Reply: We measured one device for each twist angle. This is because it is nearly impossible to obtain two devices with the exact same twist angle. On each device, we analysed several data sets acquired with separately calibrated STM tips at up to three different regions. We find the same QPI pattern for all measurements on any given device.

3. For the 30-degree twist angle sample, does the QPI ring always vanish for various scanning bias voltages?

Reply: Like for all our devices, we analysed the QPI pattern obtained at different bias voltages, and none of them shows any sign of the backscattering ring in the 30-degree twist angle device.

4. Could the authors add the pixel number and scanning speed information for the $dI/dV(V)$ maps used for the QPI study?

Reply: As suggested by the referee, we have added the information for the $dI/dV(V)$ maps in the method of the revised manuscript by replacing

~~$dI/dV(V, \vec{r})$ maps were measured using a standard lock-in technique with a bias modulation amplitude of $V_{\text{rms}} = 2$ mV at a frequency $f = 429$ Hz.~~

with

The $dI/dV(V, \vec{r})$ maps were measured using a standard lock-in technique with a bias modulation amplitude $V_{\text{rms}} = 2$ mV at a frequency $f = 429$ Hz. The $dI/dV(V, \vec{r})$ map in Fig. 3a has 1848×1848 pixels and was acquired with a scanning speed of 21 nm/s. The other $dI/dV(V, \vec{r})$ maps have 256×256 pixels and were acquired at a scanning speed of 20 nm/s.

5. If the authors have also done QPI measurements on small twist angle graphene/hBN samples, it would be nice for the authors to add such data to the manuscript to demonstrate that, in this case, the QPI power spectra follow the theoretically calculated QPI power spectra for graphene with a mass gap as shown in Fig. 5h.

Reply: We thank Referee#2 for this suggestion. However, we have not yet prepared any graphene/hBN device with a small twist angle. We are equally interested in measuring the QPI profile of such a device to check whether we would find a backscattering ring matching the prediction in Fig. 5h. This is work in progress.

Reviewers' Comments:

Reviewer #1:

Remarks to the Author:

The authors have answered my questions and remarks satisfactorily.

I can recommend the publication of the manuscript.

Reviewer #2:

Remarks to the Author:

The authors have responded satisfactorily to all the queries raised by this Reviewer. I can now recommend the manuscript for publication in Nature Communications.

Reviewer #3:

Remarks to the Author:

The authors have addressed my previous comments satisfactorily, so I recommend the publication of this manuscript.